# Physician and patient concordance in reporting of appropriateness and prioritization for cataract surgery

**Matthew B. Schlenker**[1,2,3]\*, **Saba Samet**[1,2], **Morgan Lim**[3,4], **Chelsea D'Silva**[3], **Robert J. Reid**[3,4], **Iqbal Ike K. Ahmed**[1,2]

**1** Department of Ophthalmology and Vision Sciences, University of Toronto, Toronto, Ontario, Canada,
**2** Prism Eye Institute, Oakville, Ontario, Canada, **3** Institute for Better Health, Trillium Health Partners,
Mississauga, Ontario, Canada, **4** University of Toronto, Toronto, Ontario, Canada

\* matt.schlenker@gmail.com

**Data Availability Statement:** All relevant data are within the paper and its Supporting information files.

## Abstract

### Background/Aims

Determine the association between physician-deemed and patient-reported appropriateness and prioritization for cataract surgery.

### Methods

Prospective cohort study of 471 patients of 7 ophthalmologists referred for cataract surgery. Ophthalmologists rated patients for cataract surgery appropriateness and prioritization using a visual analogue scale of 0–10 preoperatively. All patients completed the eCAPS Quality of Life (QoL), while 313 completed the Catquest-9SF and EQ-5D questionnaires. Regression analyses were applied to determine demographic, clinical and patient-reported outcome measures (PROMs) associated with appropriateness and prioritization.

### Results

Two clinical factors (study eye and fellow eye best-corrected visual acuity (BCVA)), 2 eCAPS (night driving difficulty, ability to take care of local errands), and 2 Catquest-9SF PROMs (recognizing faces, seeing to walk on uneven ground) were associated with appropriateness. In multivariable regression, the rating physician, 2 clinical criteria (study eye BCVA, anticipated postoperative BCVA) and 1 eCAPS QoL (night driving difficulty) were associated with appropriateness. Prioritization was associated with low income, 8 clinical criteria, 9 eCAPS, 5 Catquest-9SF, and 1 EQ-5D PROMs. In multivariable regression, 1 clinical criterion (study eye BCVA), 2 eCAPS QoL (night driving difficulty, ability to take care of local errands), and 2 Catquest-9SF PROMs (seeing prices, seeing to walk on uneven ground) were significantly associated.

### Conclusions

The eCAPS and Catquest-9SF questionnaires show some concordance with physician-deemed appropriateness, and more with prioritization. Binary conversions of PROM scales

**Funding:** The authors received no specific funding for this work.

**Competing interests:** I have read the journal's policy and the authors of this manuscript have the following competing interests: Matthew B. Schlenker: honoraria from, and consultant to, Alcon (Fort Worth, TX, USA); honoraria from, and consultant to, Allergan plc (Irvine, CA, USA); consultant to Light Matter Interaction (Toronto, ON, CA); consultant to Santen Pharmaceutical Co, Ltd (Kita-ku, Osaka, Japan); honoraria from Aequus (Vancouver, BC, CA); honoraria from Johnson & Johnson (Jacksonville, FL, USA); honoraria from Thea-Labtician (Oakville, ON, CA). Saba Samet: no financial disclosures. Morgan Lim: no financial disclosures. Chelsea D'Silva: no financial disclosures. Robert Reid: no financial disclosures. Iqbal Ike K. Ahmed: speaker's honoraria and research support from, and consultant to, Abbott Medical Optics (Abbott Park, IL, USA); consultant to Acucela (Seattle, WA, USA); research support from, and consultant to, Aerie Pharmaceuticals (Durham, NC, USA); speaker's honoraria and research grant support from, and consultant to, Alcon (Fort Worth, TX, USA); speaker's honoraria and research support from, and consultant to, Allergan plc (Irvine, CA, USA); consultant to ArcScan (Golden, CO, USA); consultant to Bausch and Lomb (Rochester, NY, USA); speaker's honoraria and research support from, and consultant to, Carl Zeiss Meditec (Jena, Germany); consultant to Centervue (Padova, Italy); consultant to Clarity Medical Systems (Pleasanton, CA, USA); consultant to ElutiMed (New Orleans, LA, USA); consultant to Envisia Therapeutics (Morrisville, NC, USA); consultant to Equinox (Newport Beach, CA, USA); consultant to Eyelight (Funo, Italy); consultant to ForSight Labs (Menlo Park, CA, USA); research support from, and consultant to, Glaukos (San Clemente, CA, USA); consultant to Gore (Newark, DE, USA); consultant to InjectSense (Emeryville, CA, USA); consultant to Iridex (Mountain View, CA, USA); consultant to iStar (Wavre, Belgium); research support from, and consultant to, Ivantis (Irvine, CA, USA); consultant to KeLoTec (Orange County, CA, USA); consultant to LayerBio (Medford, MA, USA); consultant to Leica Microsystems (Wetzlar, Germany); research support from, and consultant to, New World Medical (Rancho Cucamonga, CA, USA); consultant to Omega Ophthalmics (Lexington, KY, USA); consultant to Ono Pharma (Chuo-ku, Osaka, Japan) consultant to PolyActiva (Melbourne, VIC, Australia); consultant to Sanoculis (Kiryat Ono, Israel); consultant to Santen Pharmaceutical Co, Ltd (Kita-ku, Osaka, Japan); consultant to Science Based Health (Spring, TX, USA); consultant to

provide similar modelling, with minimal loss of explanatory power. As physician-deemed appropriateness and prioritization do not completely capture the patient perspective, PROMs may have a role in cataract surgery decision-making frameworks.

## Introduction

Age-related cataracts are a the major cause of reversible vision impairment, affecting 95 million people globally and 2.5 million in Canada [1–3]. The impact of vision decline can lead to inability to work, to care for others or to live independently, compromising patient quality of life [4–6]. Surgical management is effective, relatively low risk, with noticeably favourable outcomes, however some patients who may potentially gain the greatest benefit in quality of life continue to have difficulty undergoing this procedure in a timely fashion [4, 5].

Within Canada there is variation in wait times across provinces and specialists. In 2005, Canada's government agreed to a set of benchmarks for medically necessary treatment, with 16 weeks being the cataract surgery benchmark (Ontario Ministry of Health and Long Term Care, 2005). The current wait time however exceeds this limit with 20.6 weeks of national median wait time, at a range of 12.0 to 64.0 weeks across provinces [7]. Internationally, the median wait time for cataract surgery was less than 50 days in Italy, Hungary, Denmark, and Sweden, in 2017. Estonia and Poland have longer wait times, at a median of about seven months and over a year, respectively. Over the past decade, wait times have increased in some countries, including Canada and Portugal, while waits decreased in Spain, and remained stable in New Zealand [8]. Furthermore, the number of individuals experiencing vision impairment from cataracts is expected to increase with the aging population [3, 9]. With respect to patient outcomes, prolonged wait times has been shown to result in patient consequences including vision loss, increased rate of falls and fractures, loss of driver's license, depression, and overall decreased quality of life [4, 5]. In the context of scarce resources, there is a need for a framework to determine appropriateness and priority for patients being evaluated for cataract surgery.

Within the current framework, for non-urgent situations, patients are generally booked on a first-come, first-served basis. Clinical factors or patient-reported outcome measures (PROMs) applicable to vision impairment are not systemically part of the prioritization process. A lack of standardized tools to determine firstly appropriateness for surgery, and secondly prioritization in waitlist placement, results in those experiencing minimal impact on daily life potentially taking precedence over those having more disability. The development of clinically useful appropriateness and prioritization metrics will allow for better directed resources according to patient needs, along with timelier access to surgery.

The idea of redesigning the model of care for cataract surgery has been explored worldwide. The United States developed one of the first appropriateness tools in 1990, however applicability to modern day is limited with the advent of phacoemulsification [10]. A number of other tools have been developed in Spain, Korea, Sweden, New Zealand and Canada [11–17]. The Western Canada Waiting List Project (WCWL) tool's psychometric properties support evidence of inter-rater and test-retest reliability [16], as well as for convergent and predictive validity [18]. Other tools have either not been validated, show weak correlation between prioritization scoring and PROMs, or have not been evaluated in clinical practice [15, 19].

The next question is how to determine impairment. We can ask the physician, who has clinical acumen, context, and objective/measurable metrics; the physician's perspective is

SOLX (Waltham, MA, USA); consultant to Stroma (Irvine, CA, USA); consultant to TrueVision (Santa Barbara, CA, USA). This does not alter our adherence to PLOS ONE policies on sharing data and materials.

particularly important for those who lack insight into the severity of their symptoms or who are unable to vocalize their functional loss. We can also ask the patient, who can provide their perspective on a day-to-day impact of their symptoms, which may or may not be captured by the physician's in-office clinical assessment (e.g., high contrast visual acuity measurement may not reflect visual status in day-to-day settings).

The aim of this study is to determine what factors into physician-deemed appropriateness and prioritization for cataract surgery, and how well their rankings correlate with PROM criteria. To explore these parameters, a number of clinical criteria and PROMs were investigated, namely the electronic cataract appropriateness and prioritization system (eCAPs) recently developed by our group (S1 Fig), the visual functioning Catquest-9SF questionnaire (S2 Fig), and the more generic health-related quality of life EQ-5D questionnaire (S3 Fig). By determining the specific clinical criteria and PROMs that are best associated with physician-assigned appropriateness and prioritization, we can establish which clinical factors physicians most highly weigh in their appropriateness and prioritization rankings, and understand how the physician's perspective relates to the patient's perspective. With this baseline in place, we can then explore to what degree these factors are improved by surgery, and use this information to develop a more efficient, patient-centered surgical system.

## Methods

### Participants

Participants were recruited from the practice of 7 surgeons across 2 sites, at their preoperative appointment or over the phone to ensure recruitment from those unable to be reached in person or those who did not have a preoperative appointment scheduled. The protocol adhered to the tenets of the declaration of Helsinki and Trillium Health Partners Research Ethics Board approval was obtained. Patients provided written informed consent prior to study participation. Inclusion criteria included: 1) patients referred for cataract surgery consultation, 2) 40 to 85 years of age, 3) planned one eye surgery, 4) no previous surgery except for previously implanted monofocal lens in the contralateral eye at a minimum of 4 weeks prior to recruitment, 5) patients undergoing both advanced IOL and ultrasound testing, 6) able to read and understand English. Patients were excluded based on the following criteria: 1) previously implanted multifocal lens in the contralateral eye, 2) repeat cataract surgery in the study eye, 3) presence of chronic pain related condition, 4) use of chronic pain medication, 5) clinically incompetent by the Short Orientation Memory Concentration Test (SOMC). Patients over 85 were excluded based on 1) the higher risks of systemic complications during surgery with older age, 2) the higher prevalence of pre-existing ocular diseases which may affect cataract surgery outcomes in elderly patients, and 3) the association of increasing age with poorer cataract surgery outcomes [20]. Patients with previous multifocal lenses were excluded to prevent the possibility of visual disturbances and dysphotopsias from the lens itself contributing to responses on glare and night driving difficulty. Regardless, the number of patients receiving multifocal lenses in this population was small.

### Patient reported outcome measures (PROMs)

The electronic cataract appropriateness and prioritization system (eCAPs), is a modification of the WCWL [21]. eCAPS provides a set of 10 clinical criteria scored by the surgeon and 12 quality of life measures rated on 3 levels by the patient. The Catquest-9SF is a Rasch-scaled, visual functioning questionnaire, consisting of two global items and seven difficulty items on a 5-point scale, with excellent psychometric qualities of validity and responsiveness [22–24]. The EQ-5D is a generic health-related quality of life questionnaire utilized and validated in many

countries, which assesses five dimensions of health including mobility, self-care, usual activities, pain/discomfort and anxiety/depression on a 5-point scale, and provides a visual analogue scale (VAS) from 0 to 100 to rank perceived overall health state [25]. Participants completed the eCAPS quality of life (QoL), the Catquest-9SF, as well as the EQ-5D questionnaires, prior to assessment by their ophthalmologist (questionnaires available in S1–S3 Tables).

## Surgeon-collected data

At time of consultation, surgeons obtained best-corrected Snellen visual acuity (BCVA) and completed the eCAPS clinical criteria for each participant. For those recruited over the phone, surgeons completed a chart review to complete clinical criteria parameters, with the patient's most recent BCVA also extracted. Appropriateness and prioritization for surgery were each rated preoperatively by the ophthalmologist on a visual analogue scale of 0 to 10. For appropriateness, a score of 0 was indicative of the least appropriate, with 10 being highly appropriate for surgery. For prioritization, a score of 0 was indicative of the lowest priority, with 10 being high priority for surgery.

## Data analysis

Descriptive statistics were applied to assess the patient population and distribution of appropriateness and prioritization scoring among physicians. The Shapiro-Wilk test was applied to test for normality. Difference in scoring patterns among surgeons was analyzed with parametric and nonparametric tests as suitable, and data Z-transformed in instances of high inter-surgeon variability. Logistic and linear regression analysis was used to explore factors associated with increasing appropriateness and prioritization scoring among preoperative demographics, clinical criteria and PROMs.

The eCAPS QoL questionnaire provides a severity scale of none, mild/moderate and severe. The Catquest-9SF utilizes a scale of 0–4 from 'cannot decide, 'no difficulty', 'some difficulty', to 'great difficulty' and 'very great difficulty', while the EQ-5D is ranked from 1 to 5 from 'no difficulty' to 'extreme/unable'. To investigate whether a simplified, binary scale more conducive to the clinical setting provides similar associations to appropriateness and prioritization, PROM responses were collapsed into dichotomous categorizations of either no difficulty or presence of difficulty, and reanalyzed.

Multivariable regression analysis (for models of appropriateness and prioritization) was performed for each questionnaire. Statistical significance was set at $p \leq 0.05$. All statistical tests were performed using SPSS software (IBM SPSS Statistics for Macintosh, Version 25.0. Armonk, NY, USA) and RStudio (version 1.2.1335). Simultaneous testing of the models were performed through a global $F$-test prior to follow-up t-testing for predictors, to provide some protection against type 1 error inflation from multiple comparisons.

## Results

A total of 471 patients from 7 participating ophthalmologists were recruited preoperatively. 47.1% of consultations were done for the left eye, 49.0% for the right, and 3.8% were unidentified. All participants completed the eCAPS QoL questionnaire with surgeons completing the corresponding eCAPS Clinical Criteria; 313 of these patients also completed the Catquest-9SF and EQ-5D questionnaires. Patient age ranged from 42 to 85 years, 55% were females. Most patients (40%) were of European ethnicity, while 28.6% and 27.8% were of Asian and American (North, Central, South American) backgrounds, respectively. A variety of educational and financial backgrounds were reported, with approximately equal proportions of those with

<$30 000 annual income to those above $70 000, along with participants who had less than high school level education to those with a university degree (Table 1).

86.2% of patients had a cataract in the fellow eye at time of consultation, and 34.2% of patients were on the waitlist for cataract surgery in the fellow eye. In the study eye, mean BCVA was 0.55 ± 0.50 logMAR (20/70), median 0.40 logMAR (0.30 IQR). In the contralateral eye, patients had a mean BCVA of 0.32 ± 0.35 logMAR (20/40), median 0.30 logMAR (0.22 IQR). The majority of patients had BCVA of 20/40–20/150 in the study eye with BCVA of 20/30 or better in the contralateral eye (Fig 1).

Fig 2 illustrates distributions of clinical criteria and questionnaire ratings. Among clinical criteria, most patients had good anticipated visual outcome (Fig 2A; C.1). The majority of patients did not have anisometropia (C.2), were not monocular (C.3), had routine case complexity (C.4), did not have ocular comorbidities (C.5), comorbidity impact on postoperative improvement or timing of surgery (C.6, C.7), or cataract impact on comorbidity treatment (C.8). Within eCAPS QoL parameters, a large proportion were found to experience glare (Q.1; 66.2%) and night driving difficulty (Q.2; 53.2%). Extent of impairment in visual function (Q.3) was mostly of mild/moderate severity with the majority of patients having minimal difficulty with the remaining QoL parameters. The Catquest-9SF showed a wider distribution in ratings with most patients selecting the 'no difficulty' or 'some difficulty' severity levels. Among all factors, only satisfaction with present vision (B) had a high median rating of 'fairly dissatisfied'. None of the participants selected the 'cannot decide' option for any Catquest parameter. Median rating of all EQ-5D parameters showed patients largely had no problems with mobility (Q.1), self-care (Q.2), usually activities (Q.3), pain/discomfort (Q.4), or anxiety/depression (Q.5), with few experiencing slight to moderate problems in these areas.

**Table 1. Patient demographics.**

| Characteristic | Cohort (N = 471) |
|---|---|
| **Age** | |
| Median (range)—yr | 70 (42–85) |
| **Gender**—no. (%) | |
| Female | 259 (55) |
| Male | 212 (45) |
| **Ethnicity**—no. (%) | **N = 468** |
| Africa | 17 (3.6) |
| Americas (North, Central, South) | 130 (27.8) |
| Asia | 134 (28.6) |
| Europe | 187 (40.0) |
| **Household Income**—no. (%) | **N = 311** |
| < $30 000 | 80 (25.7) |
| $30 000–$49 999 | 78 (25.1) |
| $50 000–$69 999 | 50 (16.1) |
| $70 000 + | 103 (33.1) |
| **Education**—no. (%) | **N = 467** |
| Lower than High School | 76 (16.3) |
| High School | 119 (25.5) |
| Apprenticeship | 26 (5.6) |
| College | 86 (18.4) |
| University | 160 (34.3) |

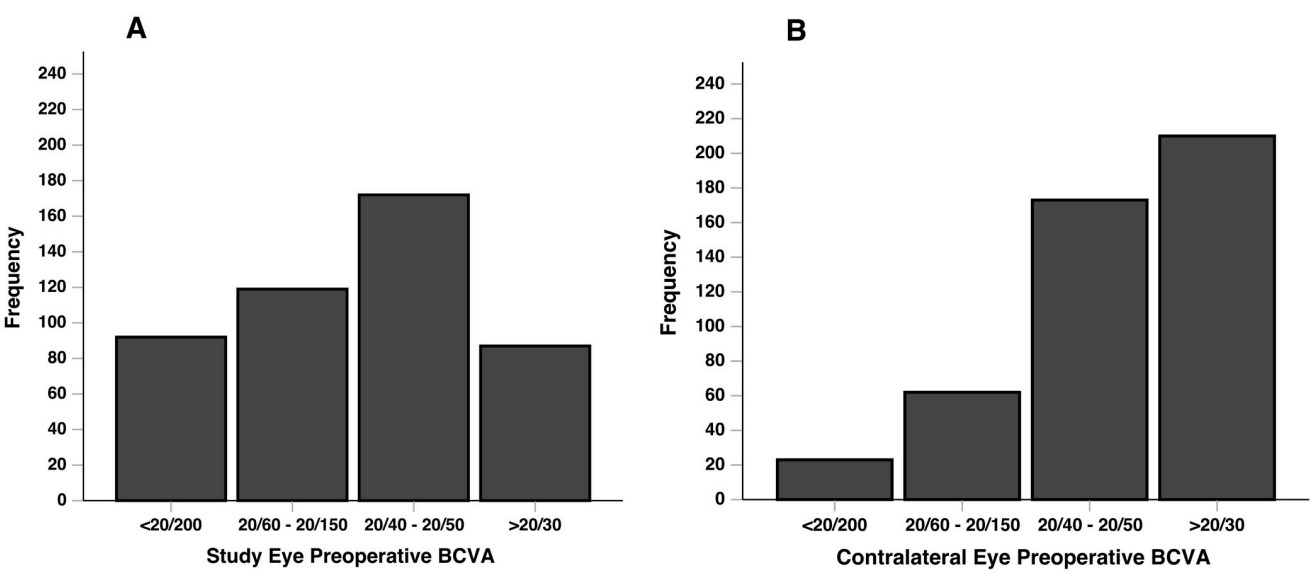

**Fig 1. Study eye (A) and contralateral eye (B) preoperative best-corrected visual acuity (BCVA).**

## Appropriateness

Physician appropriateness scores demonstrate a left skew, where the majority of patients ranked 7 or higher (81.3%) with a steep decline in frequency for those ranked 6 or lower (Fig 3A). Given this distribution, suggesting that physicians reported this measure as a dichotomous outcome, a threshold of 7/10 was utilized to convert appropriateness to a binary high versus low metric. Chi-square test however showed a statistically significant association between the rating physician and scoring of high or low appropriateness at a threshold of $\geq$7/ 10 ($p < 0.001$). Applying a Z-transformation to raw appropriateness scores for each surgeon, the resulting distribution did not demonstrate normality ($W(459) = 0.893$, $p < 0.001$ for all scores combined, $p < 0.001$ for each physician), having a left skew with the majority of Z-transformed scores above -0.65. Dividing scoring to high versus low appropriateness at the -0.65-threshold continued to show heterogeneity in physician ratings, with a significant correlation between reporting physician and appropriateness ($p = 0.036$). As the distribution of raw scores for each surgeon, despite being heterogenous among raters, showed a division at appropriateness score of 7, the 7/10 threshold of high versus low appropriateness was utilized for subsequent analyses, and the rating physician was controlled for in each regression analysis.

Using logistic regression analysis, appropriateness ranking was found to be strongly associated with surgical booking (OR 11.33 (95% CI, 4.33–29.63), $p < 0.001$ for $\geq$7 appropriateness cut-off), however 55.3% of those not scheduled for surgery were still considered potentially appropriate. Furthermore, every point increase in priority scoring was associated with a 1.87 (95% CI, 1.59–2.20) time increase in odds of being considered highly appropriate (Fig 3C; $p < 0.001$, $R^2 = 25.2\%$).

Logistic regression analysis was applied to assess for concordance between physician-determined appropriateness ranking and each of demographics, preoperative clinical criteria and PROMs.

**Demographics.** As outlined in S1 Table, no demographic factors were significantly associated. The specific physician deeming the patient as high versus low appropriateness was a major factor, explaining 19.1% of the model variation. Age, gender, ethnicity, annual household income, or education were not significantly concordant.

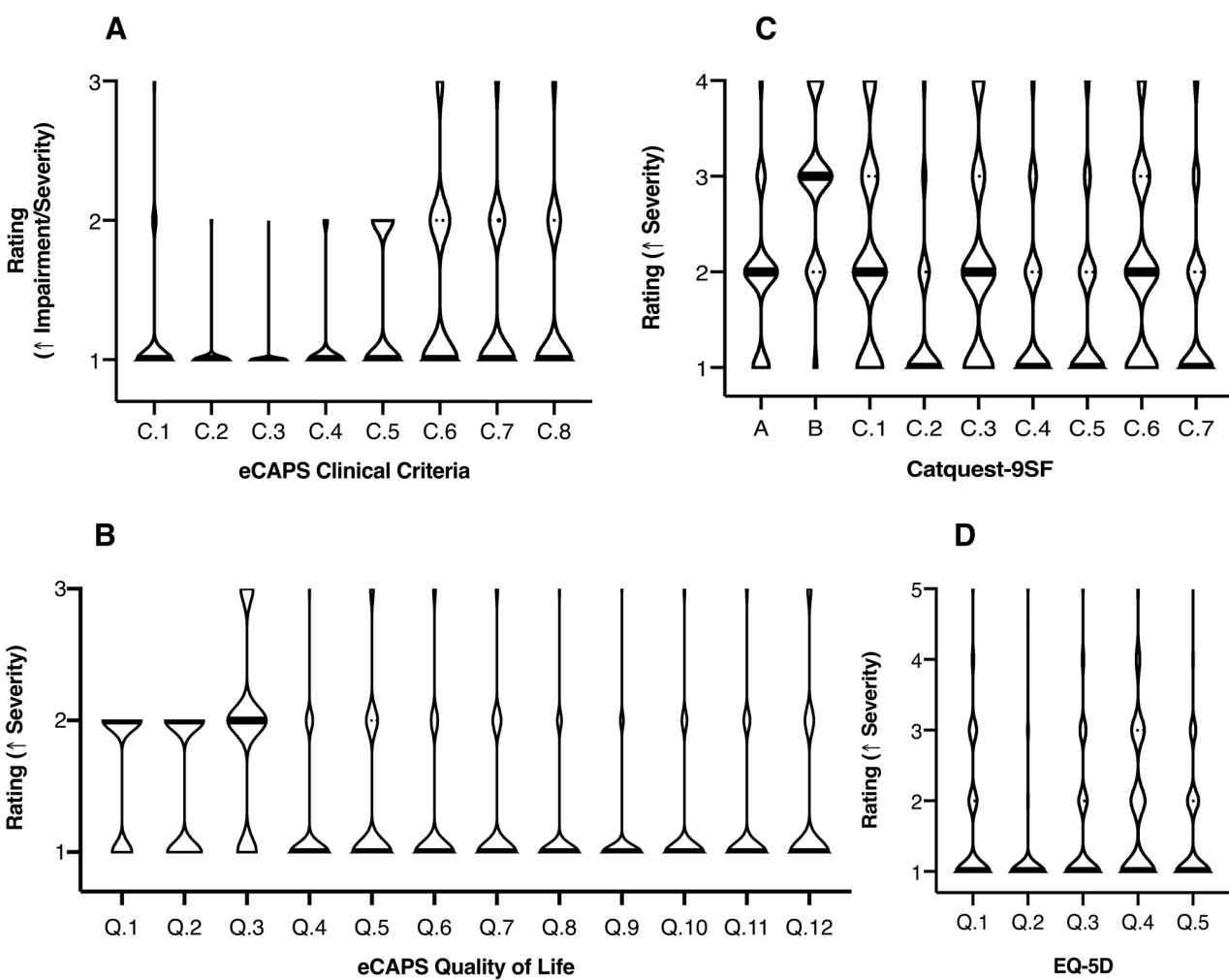

**Fig 2. Violin plots of clinical criteria and questionnaire ratings.** Median plotted as solid thick lines, with quartiles as dotted lines. (A) eCAPS Clinical Criteria; (B) eCAPS Quality of Life; (C) Catquest-9SF; (D) EQ-5D.

**eCAPS clinical criteria.** Assessing clinical factors and their association with a high rating of appropriateness for surgery at a threshold of 7/10, the significant parameters were visual acuity of the study eye (OR 2.40 (95% CI 1.21–4.75) for those with BCVA 20/40–20/50, OR 5.34 (95% CI 2.41–11.84) for BCVA 20/60–20/150, and OR 7.74 (95% CI 3.08–19.44) for BCVA ≤20/200), and acuity of the fellow eye, having a similar incremental increase in odds ratio with worsening vision (S2 Table). Anticipated postoperative BCVA, anisometropia, monocularity, case complexity, presence of ocular comorbidity, and impact on or effect of comorbidities were not significantly associated.

**PROMs.** *eCAPS*. The binary logistic regressions of PROMs and appropriateness are shown in S3 Table. Considering significance of eCAPS QoL criteria on appropriateness ranking of 7/10 or above, the questions found to be associated were night driving difficulty (OR 2.02 (95% CI 1.20–3.40)) and ability to take care of local errands (OR 2.43 (95% CI 1.07–5.52) for mild/moderate difficulty; OR 1.74 (95% CI 0.36–8.46) for severe difficulty). Analysis was done converting the scale of 1–3 to a dichotomous scale of none versus mild-severe difficulty for each parameter to simplify the approach. The dichotomous scale produced the

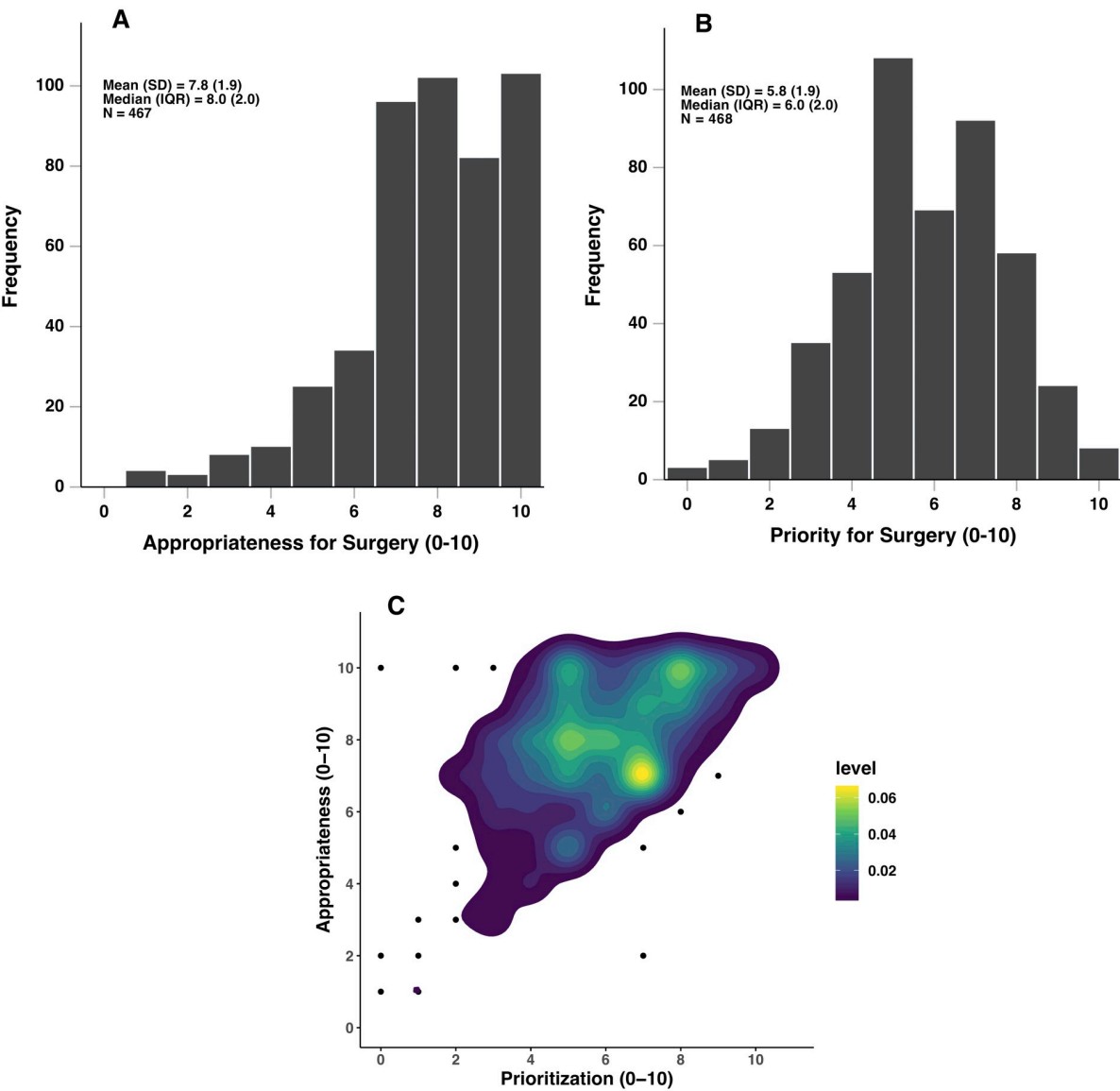

**Fig 3.** Physician ratings of appropriateness (A) and priority (B) for surgery. 2D density plot of priority versus appropriateness for surgery (C). Level bar guide defines the density of data for each of the pixels contained in the figure.

same regression results for all eCAPS parameters. The question of ability to take care of local errands was significant only at the mild/moderate level with original severity scaling; converting to a dichotomous measure, the distinction in appropriateness rating between those who ranked mild-severe is significantly different than those having no issues (OR 2.29 (95% CI 1.09–4.83)).

*Catquest-9SF.* At a 7/10 appropriateness cut-off, significant Catquest-9SF factors included recognizing faces (C.2) and difficulty seeing to walk on uneven ground (C.4). Selection of 'some difficulty' in the case of recognizing faces was associated with a 2.45 (95% CI 1.04–5.79) increase in odds of high appropriateness rating relative to no difficulty, however the binary scale did not show a significant association. Having 'some difficulty' with seeing to walk on uneven ground was also concordant (OR 2.59 (95% CI 1.15–5.87). When converted to a

dichotomous score, difficulty with seeing to walk on uneven ground was associated with a 2.31 (95% CI 1.17–4.55) increase in odds of being considered highly appropriate (S3 Table).

*EQ-5D*. The EQ-5D was similarly evaluated and found to have no questions associated with appropriateness for both the original scale and binary conversions of each parameter, as well as for reported health status and overall preoperative score.

**Multiple logistic regression.** Multiple logistic regression analysis was done to determine concordance with high appropriateness among eCAPS clinical and QoL parameters. As the dichotomized versions of the questionnaires provided similar modelling with minimal loss in explanatory power, only these converted versions of each factor were included. The three clinical questions of comorbidity impact on postoperative improvement, comorbidity impact on timing of surgery, and cataract impact on comorbidity treatment were excluded from this analysis due to limited responses to these parameters (n = 169). With this approach, significant factors from this analysis included the reporting physician, study eye preoperative BCVA, anticipated postoperative BCVA and night driving difficulty. An incremental increase in odds ratio was found with worsening study eye preoperative BCVA (OR 2.62 (95% CI 1.21–5.68) for 20/40–20/50, OR 6.01 (95% CI 2.34–15.45) for 20/60–20/150, and OR 12.37 (95% CI 4.00–38.23) for ≤20/200) as well as with better anticipated postoperative BCVA (OR 9.32 (95% CI 1.15–75.76) for questionable, and OR 13.22 (95% CI 1.72–101.55) for good expected BCVA). Night driving difficulty was associated with a 2.64 (95% CI 1.38–5.05) increase in odds of high appropriateness rating (S2 and S3 Tables).

Dichotomized versions of the Catquest-9SF parameters were also separately analyzed with multivariable logistic regression analysis, however no patient-reported outcome measures were found to be associated with appropriateness (S3 Table).

## Prioritization

Priority scoring had a more normal distribution (Fig 2B). Generally, physicians utilized the entire scale. However, both overall and individually, prioritization did not follow normality except for physician 1's scoring distribution ($W(468) = 0.970$, p < 0.001 for all scores combined, $p = 0.122$ physician 1, physician 2–7 highest $p = 0.007$). Some inter-rater variability was also seen (Table 2) and Kruskal-Wallis H test showed a statistically significant difference between physician prioritization rankings ($p < 0.001$). The prioritization distribution for each surgeon was thus Z-transformed for comparability. Subsequent to transformation, there was no significant difference between physician scores ($p = 0.982$) and no significant association between the rating surgeon and prioritization with univariable linear regression analysis ($p = 1.00$).

**Table 2. Physician appropriateness and prioritization ratings.**

| Physician | N | Appropriateness | | Prioritization | |
|---|---|---|---|---|---|
| | | Mean (SD) | Median (IQR) | Mean (SD) | Median (IQR) |
| 1 | 89 | 7.0 (2.1) | 7.0 (2.0) | 5.3 (2.2) | 5.0 (3.0) |
| 2 | 94 | 8.0 (1.8) | 8.0 (3.0) | 5.7 (1.5) | 5.0 (2.0) |
| 3 | 39 | 8.6 (1.3) | 9.0 (2.0) | 6.7 (1.7) | 7.0 (3.0) |
| 4 | 20 | 8.5 (2.5) | 9.0 (2.0) | 5.7 (2.6) | 6.0 (3.0) |
| 5 | 152 | 8.2 (1.9) | 9.0 (1.0) | 5.6 (2.2) | 5.0 (3.0) |
| 6 | 12 | 10.0 (0) | 10.0 (0) | 5.1 (0.3) | 5.0 (0) |
| 7 | 65 | 6.3 (1.2) | 7.0 (2.0) | 6.4 (1.0) | 7.0 (1.5) |

Demographics, clinical criteria and PROMs were each analyzed for concordance with Z-transformed, physician-determined prioritization ratings.

**Demographics.** All income brackets tended to have lower priority ranking than the <\$30K annual income cohort ($p = 0.023$, $R^2 = 3.1\%$). Age, gender, ethnicity, and education level however did not have a significant association (S1 Table).

**Clinical criteria.** S2 Table outlines the parameters concordant with increasing priority ranking using linear regression analysis. Significant criteria included study eye BCVA, poor visual acuity of the fellow eye, monocularity, case complexity, presence of ocular comorbidity, minimal comorbidity impact on postoperative improvement, minimal comorbidity impact on timing of surgery, moderate to severe impact of cataract on comorbidity treatment. The only factors not associated with prioritization were anticipated postoperative visual outcome and anisometropia.

**PROMs.** *eCAPS*. The QoL aggregate score (Fig 4B) as well as 9 of 12 eCAPS QoL criteria (all except for glare, night driving difficulty, and presence of other substantial disabilities), were found to have concordance with prioritization (S3 Table). Conversion to a dichotomous scale provided similar results to the original 5-point scale, except for two parameters: extent of impairment in visual function and taking part in active recreational activities, which were not significant when converted to a binary scale.

*Catquest-9SF*. 5 of 9 Catquest-9SF measures and the mean preoperative score (Fig 4C) achieved significance: difficulty in everyday life due to vision, recognizing faces, seeing prices, seeing to walk on uneven ground, and seeing to engage in an activity/hobby. Dichotomous difficulty gradings resulted in the same (in the case of recognizing faces and seeing to walk on uneven ground) or improved model significance (in the case of seeing prices and seeing to engage in an activity/hobby; although significant at specific difficulty sublevels relative to no difficulty, these two regression models were not significantly concordant with prioritization with the original 5-point scale). The fifth parameter of difficulty in everyday life due to vision, however, was associated with prioritization only with the 5-point severity level, where those with 'very great difficulty' had a significant 0.75 (95% CI, 0.12–1.38) increase in priority scoring relative to no difficulty. In the case of reading subtitles on TV, very great difficulty in this parameter had a significant association relative to no difficulty, however the overall regression model for this question did not achieve significance ($p = 0.180$).

*EQ-5D*. The EQ-5D showed poor concordance, with only 1 of the 5 parameters associated with prioritization. Those with moderate problems performing usually activities had a 0.50 (95% CI, 0.16–0.83) increase in priority scoring. The overall regression models for mobility, self-care, and pain/discomfort were not associated with prioritization ($p > 0.05$), although severe problems in each of these parameters attained significance relative to having no problems. The patient reported health status and overall EQ-5D preoperative score were also not significantly associated (Fig 4D).

**Multiple linear regression.** Among eCAPS clinical and QoL criteria, the variables found to have a significant association were study eye preoperative BCVA, night driving difficulty, and ability to take care of local errands (S2 and S3 Tables). Beta (β) weights indicate BCVA of <20/50 was the most highly associated with priority rating (β 0.90 (95% CI 0.63–1.17) for 20/60–20/150, β 0.88 (95% CI 0.59–1.17) for ≤20/200). Those with night driving difficulty and inability to take care of local errands had a less substantial but still statistically significant 0.26 (95% CI 0.08–0.45) and 0.37 (95% CI 0.13–0.62) point increase, respectively.

The Catquest-9SF consisted of two questions found to be concordant: seeing prices and seeing to walk on uneven ground (S3 Table). When including study eye preoperative BCVA in the regression however, only seeing prices maintained significance with a 0.36 (95% CI 0.07–0.66) point increase in priority scoring.

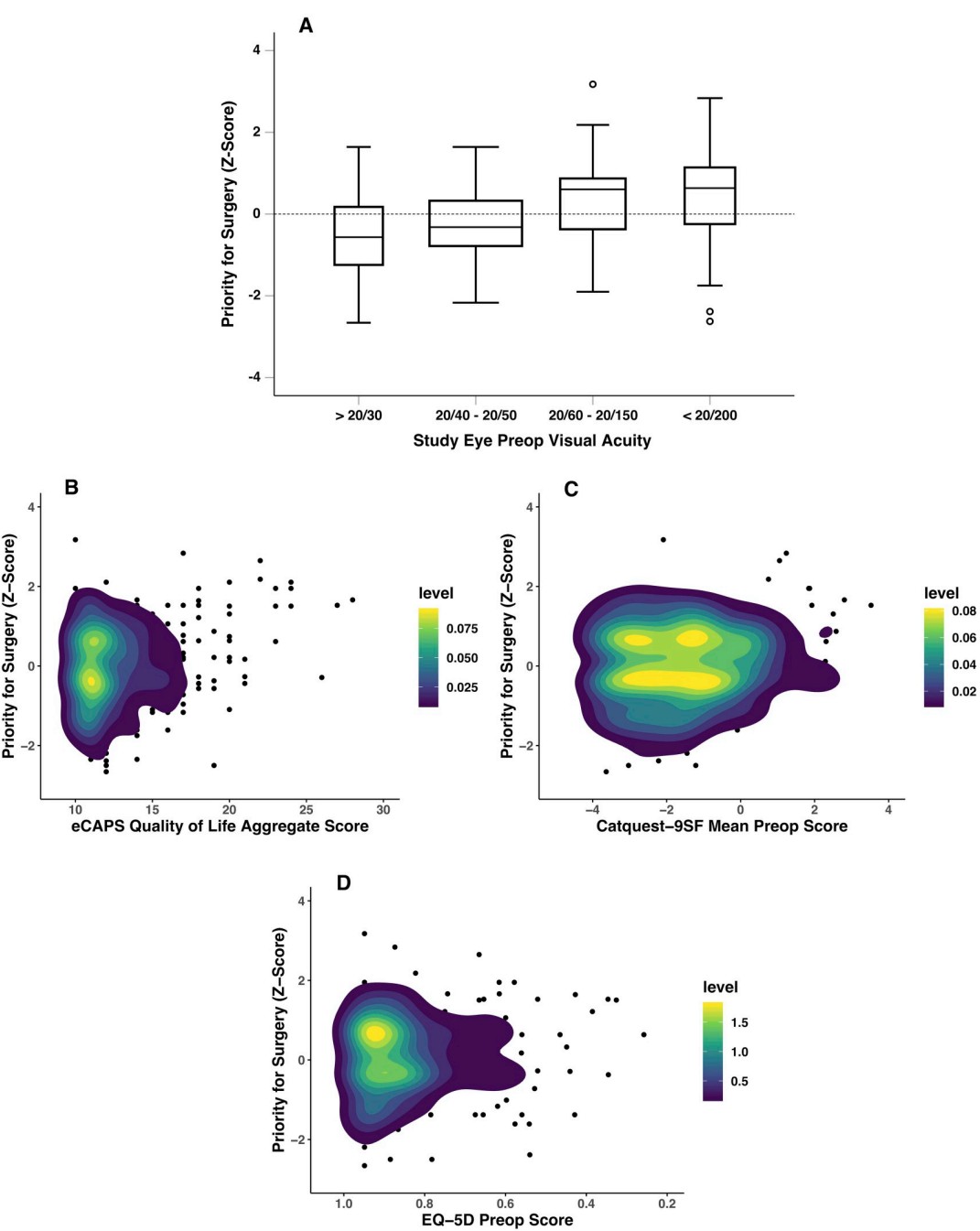

**Fig 4. Boxplot of preop visual acuity and priority ranking with box and error bar widths scaled based on count.** (A). 2D density plots of eCAPS QoL aggregate score (B), Catquest-9SF mean preop score (C), and EQ-5D score (D) versus priority for surgery. Each x-axis has been scaled to show increasing disability towards the right. Level bar guide defines the density of data for each of the pixels contained in the figure.

## Discussion

The present study evaluated the concordance between physician and patient perspectives on cataract surgery appropriateness and prioritization. Physicians evaluated clinical measures preoperatively and provided ratings for appropriateness and prioritization for surgery, while

patients ranked their functional status on both vision-specific and generic health-related questionnaires. We evaluated the degree to which these patient demographic, clinical and PROM metrics are associated with physician rankings for surgical candidacy.

With regards to appropriateness, surgeons were presented with a visual analogue scale from 0–10, however the binary-like scoring distributions implied that surgeons look at appropriateness from a binary perspective. Intuitively this is not overly surprising given the definition of appropriateness: "the quality of being suitable or proper in the circumstances" [26]. In contrast, the definition of prioritization: "the action or process of deciding the relative importance or urgency of a thing or things" implies a relative ranking [27]. As such, appropriateness analyses utilized logistic regression, accounting for the rating physician, to determine the degree to which factors are associated with high versus low appropriateness at the observed threshold of 7/10. Two clinical factors (study eye and fellow eye preoperative BCVA), 2 eCAPS (night driving difficulty and ability to take care of local errands), 2 Catquest-9SF (recognizing faces and seeing to walk on uneven ground) and 0 EQ-5D PROMs were associated with appropriateness. In multivariable regression looking at eCAPS clinical and QoL criteria, only the rating physician, 2 clinical criteria (study eye preoperative BCVA and anticipated postoperative BCVA) and 1 PROM (night driving difficulty) were found to have an association. Catquest-9SF parameters did not demonstrate any concordance in multivariable regression. For prioritization, ratings were more widely distributed with inter-surgeon variability. Using Z-score adjusted linear regression analysis, prioritization was associated with income, 8 clinical criteria, 9 eCAPS QoL, 5 Catquest-9SF, and 1 EQ-5D PROMs. The eCAPS QoL aggregate and the Catquest-9SF preoperative scores demonstrated significant associations with prioritization; the EQ-5D mean preoperative score did not. Using multivariable regression, only 1 clinical criterion (study eye preoperative BCVA) and 2 eCAPS QoL parameters (night driving difficulty, ability to take care of local errands) were significantly associated. Catquest-9SF provided 2 significant parameters (seeing prices, seeing to walk on uneven ground) but when accounting for study eye BCVA only 1 parameter remained significantly concordant (seeing prices).

We investigated whether demographic characteristics were associated with physician and patient rankings. Reassuringly, age, gender, ethnicity, and education had no significant concordance with physician ratings. The only significant factor was low income in the case of prioritization, which could be due to physicians advocating for low income patients, but could also be from other corresponding characteristics that prompted higher physician ratings or statistical artifact from multiple comparisons. Nevertheless, these demographics did not significantly alter the multivariable regression models when included for either appropriateness or prioritization.

The present study sheds light on which PROM questionnaires most strongly reflect the physician perspective. The eCAPS and Catquest-9SF showed some concordance with appropriateness and prioritization, while the EQ-5D in contrast did not. The lack of sensitivity of the EQ-5D can be due to the fact that it provides more generic parameters such as anxiety/depression, pain/discomfort, self-care, perceived overall health status, etc. which are not typically explored in the clinical setting at time of surgical decision-making, and the majority of the study population was also not experiencing difficulty in these areas at time of cataract surgery evaluation. The more vision-specific eCAPS and Catquest-9SF parameters did initially show some associations, less with appropriateness than prioritization. This may potentially be interpreted as PROMs having minimal impact on driving the decision for cataract surgery while they may be more valued when assessing where to align relative candidacy among patients. Nevertheless, when evaluated alongside other clinical criteria in multivariable regression, very few PROMs continued to demonstrate significant associations with physician-reported appropriateness and prioritization criteria. Study eye preoperative acuity demonstrated the most meaningful

concordance, with incrementally worsening vision corresponding with increasing appropriateness and priority (Fig 4A). Taken together, this data suggests that PROMs are not completely captured by physician ratings in the current surgical system.

In addition to evaluating newly developed and established quality of life questionnaires, this study explored the possibility of using dichotomized, more simplified, versions of PROM parameters. Each questionnaire provides a gradient of disability ratings; these scales were collapsed to 'no difficulty' and 'difficulty' responses and reanalyzed. The results showed that univariable regressions largely provided similar, and in some cases improved, modelling of associations when using binary conversions, with minimal loss of explanatory power. Although the specificities of disability intensity experienced by patients is lost with dichotomous scaling, these binary metrics may provide simpler and more meaningful clinical applications.

There are a number of limitations to the present study that should be noted. As discussed previously, despite being recruited from 7 ophthalmology practices across 2 sites, from both those attending appointments in person and those recruited by phone, the population of patients captured were largely of a healthy cohort. The spectrum of clinical criteria therefore tended to be of lower severity, and PROMs tended to be of minimal to moderate disability. A sample population consisting of a less homogenous cohort and a broader spectrum of quality of life responses may result in physicians more closely incorporating patient-specific metrics into surgical decision-making. Future studies may also benefit from including a larger sample size across wider geographic boundaries. Our study also demonstrated heterogeneity among physician ratings, particularly in the case of appropriateness. This may be secondary to true variations between patient populations for surgery qualification and additional corresponding variations in the surgeon impression, or due to a limitation of the 0–10 appropriateness and prioritization scales. With regards to study analysis, the issue of multiple comparisons is a limitation. One strategy used to mitigate against this has been to apply simultaneous testing of the models through a global *F*-test prior to follow-up t-testing for predictors. This provides some protection for type 1 error inflation without using an alpha adjustment procedure, which would also reduce power. Lastly, this investigation did not account for the second eye of patients, who may in actuality experience poor quality of life secondary to comorbidities in the second eye from varying pathology or may conversely compensate for disabilities stemming from the study eye using the fellow eye. Accounting for the second eye to greater detail will allow for better characterization of PROM criteria and impact on physician ratings.

In conclusion, if asked, physicians will rank patients in terms of appropriateness and prioritization, suggesting that the order of surgery would be different in such a system versus the current informal (mainly chronological) allocation system. This order does not appear to be associated with demographic characteristics such as age, gender, or education. Patients will also stratify themselves using PROM questionnaires, particularly the eCAPs and Catquest-9SF, which would also affect the order of surgery. Importantly, appropriateness and prioritization will be different if it is defined by physicians versus PROMs, and more granular PROM scaling did not improve the association. For appropriateness, the strongest correlated PROM with physician rating was night driving difficulty, and for prioritization was the ability to take care of local errands. Overall, the results suggest that assuming that both the physician and patient perspective is important during the evaluation of cataract surgery, both measures should be incorporated into an appropriateness and prioritization allocation system.

With this reference in place, future studies should aim to explore the degree to which these clinical and PROM parameters are improved postoperatively. Utilization of these surgically-modifiable metrics in a more patient-centered appropriateness and prioritization system has the potential to support streamlined, yet quality-of-life-focused cataract care.

## Supporting information

**S1 Fig. eCAPS clinical criteria and quality of life criteria.** eCAPS provides a set of 10 clinical criteria scored by the surgeon and 12 quality of life measures rated on 3 levels by the patient. (TIFF)

**S2 Fig. Catquest-9SF questionnaire.** The Catquest-9SF is a Rasch-scaled, visual functioning questionnaire, consisting of two global items and seven difficulty items on a 5-point scale. (TIFF)

**S3 Fig. EQ-5D questionnaire.** The EQ-5D is a generic health-related quality of life questionnaire, which assesses five dimensions of health including mobility, self-care, usual activities, pain/discomfort and anxiety/depression on a 5-point scale, and provides a visual analogue scale (VAS) from 0 to 100 to rank perceived overall health state. (TIFF)

**S1 Table. Binary logistic and linear regressions for demographics.** *Physician rating has been accounted for in each regression analysis for appropriateness. Statistically significant parameters are highlighted in yellow (by overall model F-test and parameter specific t-test). OR = odds ratio; CI = confidence interval. (DOCX)

**S2 Table. Logistic and linear regressions for clinical criteria.** *Physician rating has been accounted for in each regression analysis for appropriateness. Statistically significant parameters are highlighted in yellow (by overall model F-test and parameter specific t-test). eCAPS = electronic cataract appropriateness and prioritization system; BCVA = best-corrected visual acuity; OR = odds ratio; CI = confidence interval. (DOCX)

**S3 Table. Logistic and linear regressions for PROMs.** *Physician rating has been accounted for in each regression analysis for appropriateness. Statistically significant parameters are highlighted in yellow (by overall model F-test and parameter specific t-test). eCAPS = electronic cataract appropriateness and prioritization system; OR = odds ratio; CI = confidence interval; QoL = quality of life; VAS = visual analogue scale. (DOCX)

## Author Contributions

**Conceptualization:** Matthew B. Schlenker, Chelsea D'Silva, Robert J. Reid, Iqbal Ike K. Ahmed.

**Data curation:** Morgan Lim, Chelsea D'Silva.

**Formal analysis:** Matthew B. Schlenker, Saba Samet.

**Investigation:** Matthew B. Schlenker, Saba Samet.

**Methodology:** Matthew B. Schlenker, Saba Samet, Morgan Lim, Chelsea D'Silva.

**Project administration:** Matthew B. Schlenker, Morgan Lim, Chelsea D'Silva, Robert J. Reid.

**Supervision:** Matthew B. Schlenker, Robert J. Reid, Iqbal Ike K. Ahmed.

**Validation:** Matthew B. Schlenker.

**Visualization:** Matthew B. Schlenker, Saba Samet.

**Writing – original draft:** Matthew B. Schlenker, Saba Samet.

**Writing – review & editing:** Matthew B. Schlenker, Saba Samet, Chelsea D'Silva.

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
