## [Decision Letter · Decision Letter 0]

30 Mar 2021

PONE-D-21-00107

Physician and patient concordance in reporting of appropriateness and prioritization for cataract surgery

PLOS ONE

Dear Dr. Schlenker,

Thank you for submitting your manuscript to PLOS ONE. After careful consideration, we feel that it has merit but does not fully meet PLOS ONE’s publication criteria as it currently stands. Therefore, we invite you to submit a revised version of the manuscript that addresses the points raised during the review process.

We look forward to receiving your revised manuscript.

Kind regards,

Marie-Helene Errera

Academic Editor

PLOS ONE

Additional Editor Comments:

I recommend that the Authors provide the change requested by Reviewer 2, especially answering the specific questions about Statistics.

Journal Requirements:

2. Please include additional information regarding the survey or questionnaire used in the study and ensure that you have provided sufficient details that others could replicate the analyses. For instance, if you developed or modified a questionnaire as part of this study and it is not under a copyright more restrictive than CC-BY, please include a copy, in both the original language and English, as Supporting Information.

"I have read the journal's policy and the authors of this manuscript have the following competing interests:

Matthew B. Schlenker: honoraria from, and consultant to, Alcon (Fort Worth, TX, USA); honoraria from, and consultant to, Allergan plc (Irvine, CA, USA); consultant to Light Matter Interaction (Toronto, ON, CA); consultant to Santen Pharmaceutical Co, Ltd (Kita-ku, Osaka, Japan); honoraria from Aequus (Vancouver, BC, CA); honoraria from Johnson & Johnson (Jacksonville, FL, USA); honoraria from Thea-Labtician (Oakville, ON, CA).

Saba Samet: no financial disclosures.

Morgan Lim: no financial disclosures.

Chelsea D’Silva: no financial disclosures.

Robert Reid: no financial disclosures.

Iqbal Ike K. Ahmed: speaker’s honoraria and research support from, and consultant to, Abbott Medical Optics (Abbott Park, IL, USA); consultant to Acucela (Seattle, WA, USA); research support from, and consultant to, Aerie Pharmaceuticals (Durham, NC, USA); speaker’s honoraria and research grant support from, and consultant to, Alcon (Fort Worth, TX, USA); speaker’s honoraria and research support from, and consultant to, Allergan plc (Irvine, CA, USA); consultant to ArcScan (Golden, CO, USA); consultant to Bausch and Lomb (Rochester, NY, USA); speaker’s honoraria and research support from, and consultant to, Carl Zeiss Meditec (Jena, Germany); consultant to Centervue (Padova, Italy); consultant to Clarity Medical Systems (Pleasanton, CA, USA); consultant to ElutiMed (New Orleans, LA, USA); consultant to Envisia Therapeutics (Morrisville, NC, USA); consultant to Equinox (Newport Beach, CA, USA); consultant to Eyelight (Funo, Italy); consultant to ForSight Labs (Menlo Park, CA, USA); research support from, and consultant to, Glaukos (San Clemente, CA, USA); consultant to Gore (Newark, DE, USA); consultant to InjectSense (Emeryville, CA, USA); consultant to Iridex (Mountain View, CA, USA); consultant to iStar (Wavre, Belgium); research support from, and consultant to, Ivantis (Irvine, CA, USA); consultant to KeLoTec (Orange County, CA, USA); consultant to LayerBio (Medford, MA, USA); consultant to Leica Microsystems (Wetzlar, Germany); research support from, and consultant to, New World Medical (Rancho Cucamonga, CA, USA); consultant to Omega Ophthalmics (Lexington, KY, USA); consultant to Ono Pharma (Chuo-ku, Osaka, Japan) consultant to PolyActiva (Melbourne, VIC, Australia); consultant to Sanoculis (Kiryat Ono, Israel); consultant to Santen Pharmaceutical Co, Ltd (Kita-ku, Osaka, Japan); consultant to Science Based Health (Spring, TX, USA); consultant to SOLX (Waltham, MA, USA); consultant to Stroma (Irvine, CA, USA); consultant to TrueVision (Santa Barbara, CA, USA)."

Reviewers' comments:

Reviewer's Responses to Questions

**Comments to the Author**

1. Is the manuscript technically sound, and do the data support the conclusions?

Reviewer #1: Yes

Reviewer #2: Partly

2. Has the statistical analysis been performed appropriately and rigorously? 

Reviewer #1: Yes

Reviewer #2: I Don't Know

3. Have the authors made all data underlying the findings in their manuscript fully available?

Reviewer #1: Yes

Reviewer #2: Yes

4. Is the manuscript presented in an intelligible fashion and written in standard English?

Reviewer #1: Yes

Reviewer #2: Yes

5. Review Comments to the Author

Reviewer #1: Patient prioritization refers to strategic management for access to healthcare services based on a decision process that needs to be transparent and fair with reduction of errors in the selection tool. Countries with publicly funded healthcare systems experience prolonged waiting times with potential dramatic consequences. Canada having adopted a socialist type of medicine took the lead in designing priority lists such as the Cataract Priority Criteria Score (PCS), developed by the Western Canada Waiting List (WCWL) Project to determine patient prioritization for cataract surgery (by BL Conner-Spady · 2005). The Steering Committee of the Western Canada Waiting List Project adopted the following Outcome measures: Visual Function Assessment (VFA), EuroQol (EQ-5D), and best-corrected visual acuity. The current group adopted 3 kinds of questionnaires: eCAPS Quality of Life (QoL), Catquest-9SF and EQ-5. eCAPS Quality of Life (QoL), while 313 completed the Catquest-9SF and EQ-5D questionnaires. Also innovatively, they used binary conversions of PROM scales. They found in univariate analysis that Prioritization was associated with low income, 8 clinical criteria, 9 eCAPS, 5 Catquest-9SF, and 1 EQ-5D PROMs. By multivariable regression analysis, 1 clinical criterion (study eye BCVA), 2 eCAPS QoL (night driving difficulty, ability to take care of local errands), and 2 Catquest-9SF PROMs (seeing prices, seeing to walk on uneven ground) were significantly associated. The current study took prioritization to a higher level pending the future use of artificial intelligence. The paper is well written, easy to read despite the complex statistics and the different questionnaires. The authors made the paper appear concise putting the complex tables as supplements.

Reviewer #2: General comments: This observational study examines the association between physician and patient rated preoperative factors and appropriateness and prioritization for cataract surgery in a cohort of 471 patients evaluated for cataract surgery across 7 practices in Canada. The exposures of interest include physician-rated clinical factors related to the cataract based on the eCAPS clinical criteria (e.g. anticipated complexity of surgery, anisometropia, monocular status, etc.), and patient reported measures related to activities of daily living based on the eCAPS QOL and Catquest-9SF questionnaires. The outcomes include physician-rated appropriateness and prioritization for surgery, which were both rated on a scale of 0-10. The authors conclude that certain elements on the eCAPS QOL and Catquest-9SF show concordance with physician-deemed appropriateness and prioritization for cataract surgery. This study examines an interesting subject in anticipation of future resource and workforce limitations for cataract surgery, but several aspects of the study require further analysis and explanation.

Specific comments:

1) Introduction: The foundation of the authors’ justification for doing this study is that there are anticipated scarce resources for cataract surgery and that there will be a need to prioritize patients for surgery. Are there any data for current cataract surgery wait times across different healthcare settings in different countries? How about data on the consequences of delayed surgery?

2) Methods: There are a long list of inclusion and exclusion criteria. Why were patients with multifocal lenses in one eye excluded? Why were patients over age 85 excluded?

3) Methods: What variables were controlled for in the multivariable regression models?

4) Methods/Results: The authors state that physicians rated surgery appropriateness and prioritization on a scale of 0-10. Can more detail be provided regarded this rating system? Was there any standardization of the criteria for each numerical score? Furthermore, the authors decide to use a dichotomous outcome for the appropriateness measure with a binary cutoff of 7/10 for appropriate vs not appropriate. The authors state that this is because the results demonstrated a left skew where most patients were rated either above or below 7. This seems somewhat arbitrary. Were other statistical analyses considered, such as log transformation of the appropriateness score?

5) Methods/Results: The authors analyze a long list of questions from the two patient questionnaires in their regression models. Furthermore, they perform alternative analyses with different categorization of questionnaire responses. Was any consideration given to multiple comparisons? This should be accounted for in statistical analyses and/or discussion of study limitations.

6) Results/Discussion: The one consistent significant result from the patient questionnaires is that there is an association between night driving difficulty and physician-deemed appropriateness and prioritization for surgery. I am not sure how much new information this provides in terms of clinical decision-making for cataract surgery. Are there additional ways to examine the study questionnaire or results to provide a more robust system to stratify patients preoperatively for surgery?

6. PLOS authors have the option to publish the peer review history of their article (what does this mean?). If published, this will include your full peer review and any attached files.

Reviewer #1: **Yes: **Ahmad M Mansour

Reviewer #2: No

---

## [Author Response · Author response to Decision Letter 0]

17 May 2021

Please see our responses below:

Suggestion, Question, 

or Comment from the Editor

Author’s Response

Change in the Manuscript

1. “Please ensure that your manuscript meets PLOS ONE's style requirements, including those for file naming.” 

Thank you for the attached formatting guidelines.

This has been modified and marked with track changes.

2. “Please include additional information regarding the survey or questionnaire used in the study and ensure that you have provided sufficient details that others could replicate the analyses. For instance, if you developed or modified a questionnaire as part of this study and it is not under a copyright more restrictive than CC-BY, please include a copy, in both the original language and English, as Supporting Information.”

Thank you for the recommendation.

We have added the 3 questionnaires used in the study to supplemental figures (S1-S3 Figs), and cited alongside the description of each questionnaire provided in the introduction. 

The 3 questionnaires used in the study have been added to supplemental figures (S1-S3 Figs).

3. “Thank you for stating the following in the Competing Interests section. Please confirm that this does not alter your adherence to all PLOS ONE policies on sharing data and materials, by including the following statement: "This does not alter our adherence to PLOS ONE policies on sharing data and materials.”

This statement has been added to our cover letter.

4. “Please include captions for your Supporting Information files at the end of your manuscript, and update any in-text citations to match accordingly.”

Thank you for the provided formatting guidelines.

A Supporting Information section has been added to the end of the manuscript.

This section has been added to the end of the manuscript.

5. Introduction: The foundation of the authors’ justification for doing this study is that there are anticipated scarce resources for cataract surgery and that there will be a need to prioritize patients for surgery. Are there any data for current cataract surgery wait times across different healthcare settings in different countries? How about data on the consequences of delayed surgery?

We appreciate the reviewer’s provided feedback.

The data on these two points raised (wait times across different countries and consequence of delayed surgery) have been added to the introduction.

“Within Canada there is variation in wait times across provinces and specialists. In 2005, Canada’s government agreed to a set of benchmarks for medically necessary treatment, with 16 weeks being the cataract surgery benchmark (Ontario Ministry of Health and Long Term Care, 2005). The current wait time however exceeds this limit with 20.6 weeks of national median wait time, at a range of 12.0 to 64.0 weeks across provinces [7]. Internationally, the median wait time for cataract surgery was less than 50 days in Italy, Hungary, Denmark, and Sweden, in 2017. Estonia and Poland have longer wait times, at a median of about seven months and over a year, respectively. Over the past decade, wait times have increased in some countries, including Canada and Portugal, while waits decreased in Spain, and remained stable in New Zealand [8]. Furthermore, the number of individuals experiencing vision impairment from cataracts is expected to increase with the aging population [3, 9]. With respect to patient outcomes, prolonged wait times has been shown to result in patient consequences including vision loss, increased rate of falls and fractures, loss of driver’s license, depression, and overall decreased quality of life [4, 5].”

6. Methods: There are a long list of inclusion and exclusion criteria. Why were patients with multifocal lenses in one eye excluded? Why were patients over age 85 excluded?

Thank you for the questions raised regarding the study’s exclusion and inclusion criteria. The explanation for why multifocal lenses in one eye were excluded and the age restriction has been added to the methods. 

In addition to the rationale provided, our study population has generally had a relatively low number of multifocal lens implantation. As such, very few patients were excluded in this regard. 

“Patients over 85 were excluded based on 1) the higher risks of systemic complications during surgery with older age, 2) the higher prevalence of pre-existing ocular diseases which may affect cataract surgery outcomes in elderly patients, and 3) the association of increasing age with poorer cataract surgery outcomes [20]. Patients with previous multifocal lenses were excluded to prevent the possibility of visual disturbances and dysphotopsias from the lens itself contributing to responses on glare and night driving difficulty. Regardless, the number of patients receiving multifocal lenses in this population was small.”

7. Methods: What variables were controlled for in the multivariable regression models?

We thank the reviewer for the provided question on methods. As a statistically significant association was found between the rating physician and scoring of high versus low appropriateness, the rating physician was controlled for in each regression analysis within the assessment of appropriateness. For prioritization, the distribution for each surgeon was Z-transformed for comparability and no significant association was found between the rating surgeon and prioritization. Demographics including age, gender, ethnicity, annual household income, and education did not significantly alter the final regression models, and were thus excluded. A regression analysis on demographics however has been included in the manuscript for reference (S1 Table).

“As the distribution of raw scores for each surgeon, despite being heterogenous among raters, showed a division at appropriateness score of 7, the 7/10 threshold of high versus low appropriateness was utilized for subsequent analyses, and the rating physician was controlled for in each regression analysis.”

“We investigated whether demographic characteristics were associated with physician and patient rankings. Reassuringly, age, gender, ethnicity, and education had no significant concordance with physician ratings. The only significant factor was low income in the case of prioritization, which could be due to physicians advocating for low income patients, but could also be from other corresponding characteristics that prompted higher physician ratings or statistical artifact from multiple comparisons. Nevertheless, these demographics did not significantly alter the multivariable regression models when included for either appropriateness or prioritization.”

8. Methods/Results: The authors state that physicians rated surgery appropriateness and prioritization on a scale of 0-10. Can more detail be provided regarded this rating system? Was there any standardization of the criteria for each numerical score? Furthermore, the authors decide to use a dichotomous outcome for the appropriateness measure with a binary cutoff of 7/10 for appropriate vs not appropriate. The authors state that this is because the results demonstrated a left skew where most patients were rated either above or below 7. This seems somewhat arbitrary. Were other statistical analyses considered, such as log transformation of the appropriateness score?

Appropriateness and prioritization for surgery were each rated preoperatively by the ophthalmologist on a visual analogue scale of 0 to 10. 0 was deemed as not appropriate or lowest priority, while 10 was deemed as most appropriate and highest priority. Physicians provided their global assessment and there was no additional information provided to them about the scale. Not unlike if this scale were to be rolled out on a larger scale, where different physicians may utilize the scale differently.

For appropriateness rating, we used graphical display and descriptive statistics of the raw data to interpret scoring patterns and plan subsequent analyses. For this measure, we found on the histogram that most patients received high appropriateness scoring at a level of 7 or above—the scale seems to be utilized by physicians in a binary fashion. In contrast, prioritization distribution was more distributed along the provided scale (Fig 3). Given that surgeons were provided the same 0-10 scale for each measure of appropriateness and prioritization but the scales were utilized differently, we interpreted the data to mean that most patients were deemed appropriate for surgery at a score of 7 or above, while surgeons ranked priority as a more hierarchical measure along the entire scale. “Appropriateness and prioritization for surgery were each rated preoperatively by the ophthalmologist on a visual analogue scale of 0 to 10. For appropriateness, a score of 0 was indicative of the least appropriate, with 10 being highly appropriate for surgery. For prioritization, a score of 0 was indicative of the lowest priority, with 10 being high priority for surgery.”

“Physician appropriateness scores demonstrate a left skew, where the majority of patients ranked 7 or higher (81.3%) with a steep decline in frequency for those ranked 6 or lower (Fig 3A). Given this distribution, suggesting that physicians reported this measure as a dichotomous outcome, a threshold of 7/10 was utilized to convert appropriateness to a binary high versus low metric.”

9. Methods/Results: The authors analyze a long list of questions from the two patient questionnaires in their regression models. Furthermore, they perform alternative analyses with different categorization of questionnaire responses. Was any consideration given to multiple comparisons? This should be accounted for in statistical analyses and/or discussion of study limitations.

We thank the reviewer for the question regarding multiple comparisons. Multiple comparisons is a problem on any large-scale data analysis. One strategy used to mitigate against this has been to perform a simultaneous test of the models through a global F-test. This gives some protection against type 1 error inflation from multiple comparisons, without using an alpha adjustment procedure like Bonferroni correction which would also reduce power. As such, all of the significant results mentioned have been tested through a global F-test with follow-up t-testing for predictors.

This has been added to the methods, discussion and caption of relevant tables.

“Simultaneous testing of the models were performed through a global F-test prior to follow-up t-testing for predictors, to provide some protection against type 1 error inflation from multiple comparisons.”

“With regards to study analysis, the issue of multiple comparisons is a limitation. One strategy used to mitigate against this has been to apply simultaneous testing of the models through a global F-test prior to follow-up t-testing for predictors. This provides some protection for type 1 error inflation without using an alpha adjustment procedure, which would also reduce power.”

10. Results/Discussion: The one consistent significant result from the patient questionnaires is that there is an association between night driving difficulty and physician-deemed appropriateness and prioritization for surgery. I am not sure how much new information this provides in terms of clinical decision-making for cataract surgery. Are there additional ways to examine the study questionnaire or results to provide a more robust system to stratify patients preoperatively for surgery?

We thank the reviewer for the provided feedback. The purpose of this study was to identify which PROM questions most strongly reflect the physician perspective at this time. Our data has shown that in multivariable regression, the rating physician, 2 clinical criteria (study eye BCVA, anticipated postoperative BCVA) and 1 eCAPS QoL (night driving difficulty) were associated with appropriateness. Prioritization was associated with 1 clinical criterion (study eye BCVA), 2 eCAPS QoL (night driving difficulty, ability to take care of local errands), and 2 Catquest-9SF PROMs (seeing prices, seeing to walk on uneven ground) in multivariable regression. The more vision-specific eCAPS and Catquest-9SF parameters showed some associations, less with appropriateness than prioritization. The conclusion from our data is that PROMs are not completely captured by physician ratings in the current surgical system. Importantly, the questionnaires show that appropriateness and prioritization will be different if it is defined by physicians versus PROMs. As postoperative data has not been investigated in this study, a robust system for stratification cannot yet be directly postulated. The subsequent study from our group will be to assess the change in questionnaire scores from the preoperative to the postoperative time point, to determine whether this ranking outcome will be different from ranking based on the current system. This will help identify a stratification system based on postoperative outcomes.

“With this reference in place, future studies should aim to explore the degree to which these clinical and PROM parameters are improved postoperatively. Utilization of these surgically-modifiable metrics in a more patient-centered appropriateness and prioritization system has the potential to support streamlined, yet quality-of-life-focused cataract care.”

---

## [Editor Report · Decision Letter 1]

31 May 2021

Physician and patient concordance in reporting of appropriateness and prioritization for cataract surgery

PONE-D-21-00107R1

Dear Schlenker,

We’re pleased to inform you that your manuscript has been judged scientifically suitable for publication and will be formally accepted for publication once it meets all outstanding technical requirements.

Kind regards,

Marie-Helene Errera

Academic Editor

PLOS ONE
---

## [Editor Report · Acceptance letter]

17 Jun 2021

PONE-D-21-00107R1 

Physician and patient concordance in reporting of appropriateness and prioritization for cataract surgery 

Dear Dr. Schlenker:

I'm pleased to inform you that your manuscript has been deemed suitable for publication in PLOS ONE. Congratulations! Your manuscript is now with our production department. 

Kind regards, 

on behalf of

Dr. Marie-Helene Errera 

Academic Editor

PLOS ONE